# Effects of Ultralow-Tidal-Volume Ventilation under Veno-Venous Extracorporeal Membrane Oxygenation in a Porcine Model with Ventilator-Induced Lung Injury

**DOI:** 10.3390/membranes10120379

**Published:** 2020-11-28

**Authors:** Sung Yoon Lim, Young-Jae Cho, Dong Jung Kim, Jun Sung Kim, Sanghoon Jheon, Jin Haeng Chung, Jae Ho Lee

**Affiliations:** 1Division of Pulmonary and Critical Care Medicine, Department of Internal Medicine, Seoul National University College of Medicine, Seoul National University Bundang Hospital, Seongnam 13620, Korea; nucleon727@snubh.org (S.Y.L.); lungdrcho@snu.ac.kr (Y.-J.C.); 2Department of Cardiovascular and Thoracic Surgery, Seoul National University Bundang Hospital, Seongnam 13620, Korea; mdknockout@snubh.org (D.J.K.); bboloc@snubh.org (J.S.K.); jheon@snu.ac.kr (S.J.); 3Department of Pathology, Seoul National University Bundang Hospital, Seongnam 13620, Korea; jhchung@snubh.org

**Keywords:** acute respiratory distress syndrome, ventilator-induced lung injury, extracorporeal membrane oxygenation

## Abstract

Low-tidal-volume ventilation decreases mortality in acute respiratory distress syndrome (ARDS) patients. This study investigated the effects of ultralow tidal ventilation under veno-venous extracorporeal membrane oxygenator (ECMO) support in pigs with ARDS. Eight pigs were intubated and inoculated with methicillin-resistant *Staphylococcus aureus* through bronchoscopy. Ultralow tidal ventilation (3 mL/kg) under extracorporeal membrane oxygenator (ECMO) support was applied to one group and high tidal ventilation (15 mL/kg) was applied to another group to maintain comparable oxygenation for 12 h without ECMO support. Each group had similar arterial blood gas values and hemodynamic variables at baseline and during the experiment. The high-tidal-volume ventilation group showed a gradual decline in arterial oxygen levels, and repeated ANOVA showed significant differences in oxygenation change over time in the ultralow tidal ventilation group. Inflammatory cytokine levels in the bronchoalveolar lavage fluid and lung ultrasound scores were similar between two groups. Histologic analysis showed that both groups developed pneumonia after 12 h; however, the ultralow tidal ventilation group had a lower lung injury score assessed by the pathologist. We developed the first ultralow-tidal-volume ventilation porcine model under veno-venous ECMO support. The ultralow-tidal-volume ventilation strategy can mitigate mechanical ventilator-associated lung injury.

## 1. Introduction

Since the introduction of mechanical ventilation in the 1960s, it has been and widely used in clinical practice, leading to a decrease in mortality in critically ill patients. Despite the clear benefits of this therapy, alveolar overdistension due to a large tidal volume (VT) during mechanical ventilation results in a permeability-type pulmonary edema, called ventilator-induced lung injury (VILI) [1]. This can result in the worsening of hypoxemia, which can prolong mechanical ventilation, lead to multi-system organ dysfunction, and even increase mortality [1,2]. Thus, adopting a ventilator strategy that reduces alveolar overdistension is an important goal in ventilatory management [3].

Low tidal volume to prevent overdistention has been accepted as a standard ventilation strategy after a 22% reduction in mortality was observed in patients with acute respiratory distress syndrome (ARDS) whose VT had been reduced [4]. In most clinical guidelines, it is strongly recommended to apply mechanical ventilation using low tidal volumes of 6 mL/kg of predicted body weight and maintain a plateau pressure of ≤30 cm H_2_O to mitigate VILI for all patients with ARDS [5,6]. However, in some patients, low-tidal-volume ventilation may not fully protect against VILI [7,8]. Thus, reducing the VT to 3–4 mL/kg has been proposed to further minimize the risk of VILI [9], but this could lead to a significant risk of severe respiratory acidosis [10].

An extracorporeal membrane oxygenator (ECMO) can minimize the risk of acidosis by clearing carbon dioxide (CO_2_), enabling ultralow-tidal-volume ventilation that is more lung-protective than conventional low tidal ventilation [11,12,13]. Conceptually, such strategies might improve outcomes by using VT as low as 3–4 mL/kg without respiratory acidosis and hypoxemia, but there are no data demonstrating the feasibility and effectiveness of these ultra-protective strategies with ECMO. We therefore assessed the feasibility and safety of ECMO to facilitate ultra-protective ventilation in pigs with ARDS.

## 2. Materials and Methods

All experiments were performed at the Biomedical Research Laboratory at Seoul National University Bundang Hospital. The study was approved by the Institutional Animal Care and Use Committee of Seoul National University Bundang Hospital, with strict adherence to the guidelines for the Care and Use of Laboratory Animals, National Research Council (Project No. BA1503-173/015-01). A certified and licensed veterinarian was always present, and all experimenters received training in laboratory animal science prior to experimentation.

### 2.1. Animal Preperation

The study included 8 female pigs that were approximately 90 days old, with an average weight of 65 kg (range 61–70 kg). Pigs were anesthetized with intramuscular zoletil, followed by inhalation of isoflurane at a dose of 0.8% to 1.2%. The animals were intubated using a 6.5 mm cuffed endotracheal tube and were connected to the mechanical ventilator with room air using volume-controlled ventilation. The initial ventilatory settings included a positive end-expiratory pressure (PEEP) of 5 cm H_2_O, VT of 10 mL/kg, and respiratory rate (RR) of 14–17/min. Throughout the procedure, the partial pressure of carbon dioxide (PaCO_2_) was maintained at 35–45 mmHg by increasing the RR to the maximum level preceding the onset of auto-PEEP. The fraction of inspired oxygen (FIO_2_) was set at the lowest level possible while still achieving blood PO_2_ levels within 80–100 mmHg. Vancomycin 1 g was administered intravenously over 1 h prior to the procedure in all animals.

### 2.2. Induction of Lung Injury

To replicate the actual clinical setting, we generated a two-hit model of ARDS. Lung injury was induced by administration of methicillin-resistant *Staphylococcus aureus* (MRSA) and lipopolysaccharide (LPS) combined with mechanical ventilation. A total of 75 mL of 106 colony-forming units/mL of pathogenic MRSA was inoculated and evenly distributed among every lobe of each lung using a bronchoscope. Bacteria were inoculated once the animals were hemodynamically stable after sedation and mechanical ventilation. LPS (from *Salmonella typhimurium*; Sigma-Aldrich, Youngin, Korea) was dissolved in phosphate-buffered saline to a final concentration of 10 mg/mL. LPS was delivered intravenously to a final dose of 1 mg/kg.

### 2.3. Experimental Protocol

After baseline measurements, animals were randomly divided into two groups: The control group with high-tidal-volume ventilation (*n* = 4), and the ECMO group with ultralow-tidal-volume ventilation (*n* = 4) (Figure 1). In the control group, tidal volume was initially set at 12 mL/kg at 0 h, whereas in the ECMO group, tidal volume was initially set at 3–4 mL/kg with commencement of VV-ECMO to maintain partial pressure of oxygen (PaO_2_) and PaCO_2_ levels comparable to those of the control group throughout the study. The VV-ECMO procedure was performed as follows. A 24 F double-lumen catheter (NovaPort twin, Novalung, Germany) was inserted percutaneously into the right femoral vein, and ECMO was performed with the Prolonged Life Support System (Quadrox PLS, Maquet, Rastatt, Germany). The ECMO system included a circuit primed with ringer’s lactate, a centrifugal pump (ROTAFLOW Console, Maquet, Rastatt, Germany), and a heat exchanger (Heater-Cooler Unit HCU 30, Maquet, Rastatt, Germany), which maintained a temperature of 37 °C. The initial sweep gas (100% oxygen) was set at a flow rate equal to the blood flow rate (1:1) and then titrated to maintain an PaCO_2_ level within 35–45 mmHg, with a comparable RR to the control group. The blood in the circuit was drained and infused at a flow rate of 50 mL/kg/min.

### 2.4. Data Collection

Animals were continuously monitored for mechanical ventilation parameters (VT, RR, airway pressures, and FIO_2_), heart rate, blood pressure, and body temperature. Arterial blood samples and blood gas measurements were collected from the right femoral artery at baseline (0), 6, and 12 h.

#### 2.4.1. Cytokines in Bronchoalveolar Lavage Fluid (BAL)

Two 15 mL aliquots of sterile saline solution (0.9% sodium chloride) were instilled and re-aspirated through the bronchoscope channel in the right middle lobe at 0 h (before the inoculation of the MRSA suspension) and at sacrifice (12 h). Interleukin (IL)-6 levels were measured in the BAL supernatant using ELISA as instructed in porcine-specific kits (R&D Systems Inc., Minneapolis, MN, USA).

#### 2.4.2. Lung Ultrasound

Ultrasound was performed with a Philips iE33 apparatus using a 3 to 5 MHz probe (Philips ultrasound, Bothell, WA, USA) at 0, 6, and 12 h. Six areas per hemithorax were examined. On each hemithorax, three regions of interest (anterior, lateral, and posterior fields) were identified by sternum, anterior, and posterior axillary lines. Then, each field was divided into upper and lower parts, to constitute six standard areas per hemithorax. To quantify B-lines, the B-line of each segment was counted as 0, 1, 2, 3, 4, or ≥5 counts. The score for each segment was defined as follows: Counts ≤2 were given a score of 0; counts of 3 or 4 were given a score of 1; for counts ≥5 or if consolidation was present, a score of 2 was given, and scores from all 12 segments were added.

#### 2.4.3. Sacrifice and Histopathological Assessment

At the end of the 12 h study period, euthanasia was performed in the eight piglets under general anesthesia by intravenous overdose of potassium chloride (40 mL, 0.1 g/mL). Lung tissue was processed in accordance with standard methods. Analyses of the vessels (thrombosis and endothelial lesions), pleura (acute or chronic pleuritis), and lung parenchyma were performed. Lung injury was assessed by a pathologist blinded to the group assignment. The severity of lung injury was graded according to a previously described [14] scoring system based on each of the following five variables: Alveolar and interstitial edema, inflammatory infiltration, hemorrhage, and alveolar collapse. Each variable was analyzed at a magnification of 40× and 200× and scored using a 0–3 point scale. Zero represented injury in 25% of the field; 1, injury in 50% of the field; 2, injury in 75% of the field; and 3, injury in 100% of the field. The sum of the score was calculated, and the maximum possible score was 15.

#### 2.4.4. Lung Wet-to-Dry Weight Ratios

The lung wet-to-dry (W/D) weight ratio was used as an index of lung water accumulation. The wet weights were measured immediately after its excision using a precision balance. The lung tissue was then dried in an oven at a temperature of 98 °C for 48 h (or until the weights are constant) and was re-weighed to obtain the dry weight. The W/D weight ratio was calculated by dividing the wet by the dry weight as described previously [15].

### 2.5. Statistical Analysis

All data are presented as means (SD) and were compared using an independent or paired *t*-test as appropriate. Categorical variables are expressed as the number (percentage) and were compared using Pearson’s chi-square test or Fisher’s exact test. To assess the changes in outcome measures over time, repeated-measure ANOVA was used at every time point, with the baseline values as covariates. All statistical analyses were performed by using the R software, version 3.3.1, (R Foundation Inc.; http://cran.r-project.org/). P-values less than 0.05 were considered statistically significant.

## 3. Results

### 3.1. Hemodynamic Variables

Physiological and laboratory variables are shown in Table 1. After bronchial inoculation of MRSA and injection of LPS, a rapid and persistent decrease in the mean arterial pressure was observed throughout the study. No differences in biochemical data were observed over the study period between the high and ultralow-tidal-volume ventilation groups.

### 3.2. Oxygenation and Ventilation Profiles

Table 2 and Figure 2 present the PaO_2_ in arterial blood over time after initiation of mechanical ventilation. The levels of arterial oxygenation at each time point after ventilator initiation were not different between piglets in the high- and low-tidal-volume ventilation groups. No differences in PaCO_2_ were also observed over time. However, the mean PaO_2_ at 12 h appeared to decrease and increase compared to baseline values in piglets with high-tidal-volume and low-tidal-volume ventilation groups, respectively. Repeated-measure analysis with the baseline values as covariates revealed that the change in oxygenation was significantly different between the two groups (*p =* 0.047). To maintain comparable oxygenation and ventilation between both groups, minute ventilation, peak inspiratory, and dynamic compliance were significantly higher in pigs without ECMO.

### 3.3. Inflammatory Response, Lung Ultrasound, and Histopathological Findings

The measurement of IL-6 levels in the BAL at 12 h seemed to increase in the high-tidal-volume ventilation group and decrease in the ultralow-tidal-volume ventilation group compared to the baseline (Figure 3). However, the Wilcoxon signed-rank test showed no significant differences between baseline and at 12 h in both groups (*p =* 0.465, high-tidal-volume ventilation group; *p =* 0.068, ultralow-tidal-volume ventilation group).

The total lung ultrasound score increased in both groups after the inoculation of MRSA (Figure 4). At 12 h, there were no significant differences in the number of B-lines among the regions and total ultrasound score between both groups. The fresh resection specimen from lungs showed macroscopic discoloration in the dependent portion in both groups of piglets sacrificed after 12 h (Figure 5A). Microscopic examination also revealed the presence of diffuse alveolar damage with interstitial inflammatory cell infiltration in both groups. (Figure 5B). However, the lung specimens from the high tidal ventilation group showed more extensive and multifocal consolidative lesions, and a higher lung injury histology score compared to the low tidal ventilation group (Figure 5C, *p =* 0.017). The severity of lung injury in the lower lobe was comparable between both groups, whereas piglets in the high tidal ventilation had markedly severe lung injury in the upper lobe. To quantify the lung water content, the ratio of wet/dry weight was measured. The lung W/D ratios were comparable between both groups (Figure 5D).

## 4. Discussion

This experimental study showed that the use of ECMO with ultralow-tidal-volume ventilation in pigs with ARDS attenuated lung injury from mechanical ventilation and infection. The combination of MRSA inoculation and mechanical ventilation injury caused significant pulmonary inflammation. In comparison with the high-tidal-volume ventilation group, low-tidal-volume ventilation with ECMO showed comparable oxygenation and carboxylation at each time point after injury. However, the PaO_2_ gradually decreased at 12 h from the baseline in pigs with high-tidal-volume ventilation, and a repeated ANOVA test showed significant differences in oxygenation changes over time between both groups. Although the inflammatory response and lung ultrasound score were similar between both groups, histopathological examination of the lungs in the high-tidal-volume ventilation group demonstrated severe damage in comparison to the ultralow tidal ventilation group. Moreover, the lung injury score, as assessed by the pathologist, was significantly higher in the high tidal ventilation group.

A few studies have examined the feasibility and efficacy of ultralow tidal ventilation with extracorporeal life support [16,17]. Two retrospective studies investigated patients undergoing ultralow tidal ventilation with simultaneous ECMO and showed that the ventilation strategy could be safely and feasibly applied in real-world conditions [18,19]. However, the previous studies reported inconsistent results in terms of the positive effects of ultralow tidal ventilation. One study showed improved dynamic compliance on day seven after ultralow tidal ventilation with ECMO support [18], whereas the other study showed no significant differences in ventilatory parameters between survivors and nonsurvivors among patients with lung protective ventilation [19].

Combes et al. also showed that extracorporeal carbon dioxide removal can be used to minimize respiratory acidosis while applying ultra-protective ventilation (VT 4 mL/kg and plateau pressure ≤25 cmH_2_O) in ninety-five patients with moderate ARDS [12]. The proportion of patients who achieved ultra-protective ventilation by 8 and 24 h was 78% and 82%, respectively. However, they did not provide any clinical evidence for outcomes in ARDS patients, and did not use ECMO but extracorporeal carbon dioxide removal for lung protective setting, which could not be applied in the setting of severe hypoxemic respiratory failure.

In the current study, ultralow tidal ventilation with ECMO support mitigated diffuse lung damage according to the gross pathological appearance and semi-quantitative scoring system. More severe lung injury was present in the upper lobe due to high tidal ventilation. To the best of our knowledge, this is the first experimental study to provide evidence that ultralow-tidal-volume ventilation with simultaneous ECMO could alleviate microscopic lung injury induced by a mechanical ventilator. Lung ultrasound was also performed to assess the severity of lung injury in this study. Its use for evaluating the severity of lung injury has been validated not only in critically ill patients with ARDS but also in the experimental piglet model [20,21,22,23]. In this model, B-lines were correlated with increasing extravascular lung water, and it was detectable before the onset of functional impairments such as hypoxemia [23]. The total lung ultrasound score defined by the counts of B-lines was increased in both groups, showing a trend for more severe injury in the high tidal ventilation group; however, statistically significant differences were not observed in the lung injury score according to the regions and total ultrasound score between the two groups. These findings might be attributed to difficulty accessing the ventral region of animals with an ultrasound probe.

Despite the beneficial effects of protective ventilation on histopathologic findings in the lung, BAL IL-6 levels, a major proinflammatory cytokine, did not significantly differ between the ultralow tidal and high tidal ventilation group. As animals were sacrificed at 12 h after ventilator initiation, the duration might not have been sufficient to show the difference in IL-6 levels between the two groups. A significant increase in BAL IL-6 concentrations was observed at 24 h, peaking at post-inoculation day 7 in other studies [24,25]. Additionally, ECMO support might increase the level of IL-6 early in the course [26], leading to comparable levels of IL-6 between the two groups.

This study has several limitations. First, the exogenous administration of a high bacterial inoculum in a previously healthy piglet does not necessarily reflect the complexities of the development of pneumonia in humans. Secondly, we assigned animals ventilated with high tidal volume as a control group, rather than with low tidal volume. As low-tidal-volume ventilation is used in standard practice for ventilatory management in patients with acute respiratory failure, a control group with low tidal ventilation might be better to demonstrate the effect of ultralow tidal ventilation with the use of ECMO. However, high-tidal-volume ventilation was inevitable in the control group to maintain comparable oxygenation and ventilation with the experimental group with ECMO. Third, animals were observed over a relatively short period of time, which precludes the long-term effects of ultralow tidal ventilation. However, the beneficial effect of ultralow tidal ventilation in the lung pathology was observed within 12 h of injury in our study. Additionally, the arterial oxygen levels of pigs without ECMO support was gradually decreased and the animals could not survive longer than the 12 h due to severe hypoxemia and subsequent lactic acidosis. Finally, the low number of animals used could lead to type II errors, attenuating the validity of our findings. Nevertheless, these limitations do not affect our positive findings. In conclusion, the present study demonstrated that ECMO with ultralow-tidal-volume ventilation in pigs with ARDS was feasible and facilitated lung protection from VILI.

## Figures and Tables

**Figure 1 membranes-10-00379-f001:**
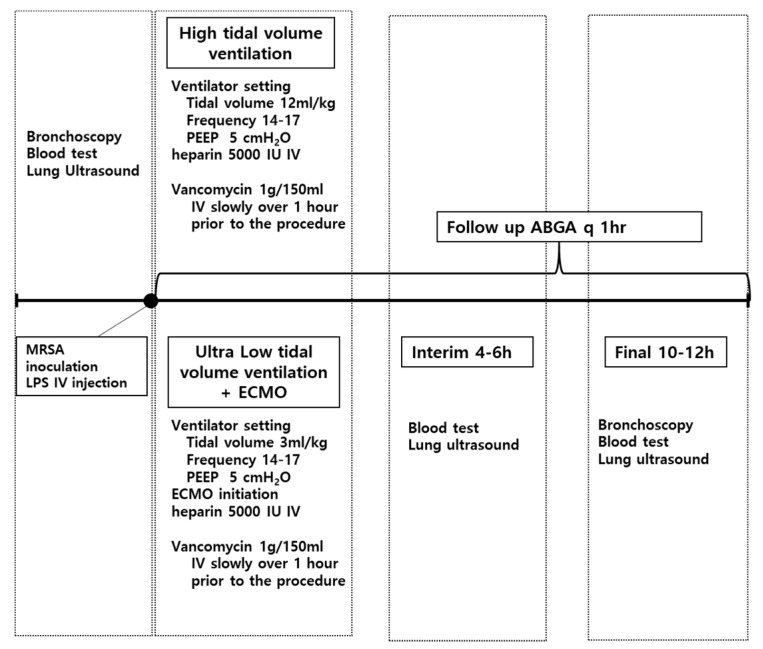
Schematic representation of the experiment protocol. MRSA: Methicillin-resistant *Staphylococcus aureus*, LPS: Lipopolysaccharide, IV: Intravenous, PEEP: Positive end-expiratory pressure, ECMO: Extracorporeal membrane oxygenation, ABGA: Arterial blood gas analysis, h: Hour.

**Figure 2 membranes-10-00379-f002:**
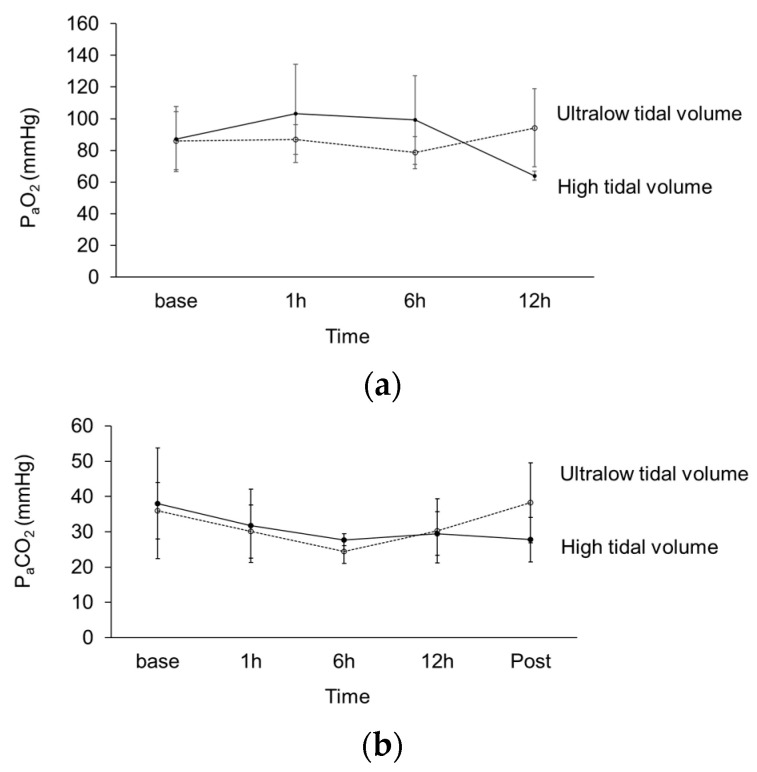
Change in the arterial partial pressure of (**a**) oxygen (PaO_2_) and (**b**) carbon dioxide (PaCO_2_), with the straight line depicting the high-tidal-volume ventilation group and the dotted line depicting the ultralow tidal ventilation group under ECMO support; h: Hour.

**Figure 3 membranes-10-00379-f003:**
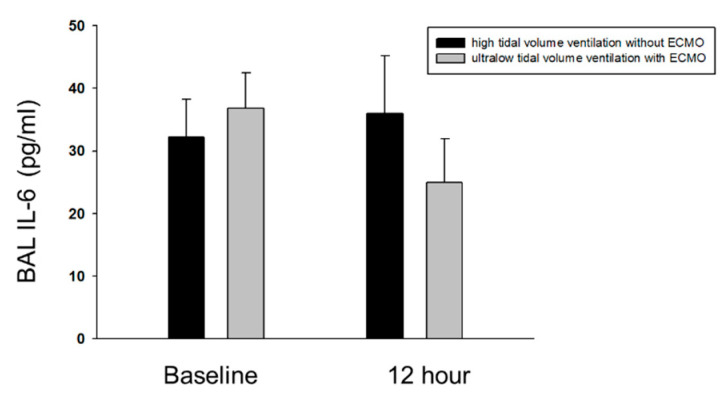
Comparison of IL-6 cytokine in bronchoalveolar lavage fluid; BAL: Bronchoalveolar lavage fluid, ECMO: Extracorporeal membrane oxygenation.

**Figure 4 membranes-10-00379-f004:**
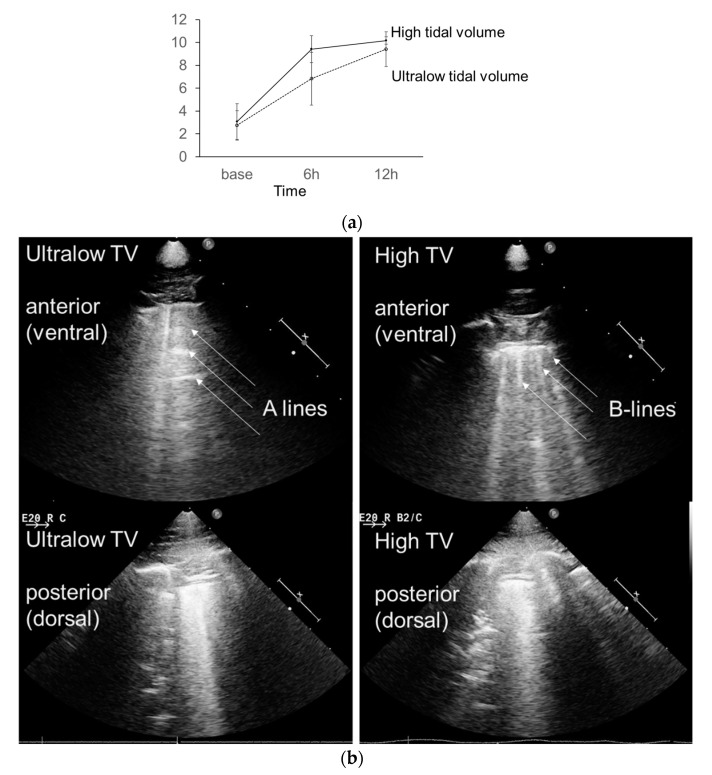
(**a**) Changes in lung ultrasound score, with the straight line depicting the high-tidal-volume ventilation group and the dotted line depicting the ultralow tidal ventilation group under ECMO support. (**b**) The lung ultrasound shows A lines (upper left panel) and typical B lines (upper right panel) in the anterior fields of the right hemithorax, with and without ECMO support, respectively. The lung ultrasound shows multiple B-lines with consolidation (lower panel) in the posterior fields of the right hemithorax of both groups; h: Hour.

**Figure 5 membranes-10-00379-f005:**
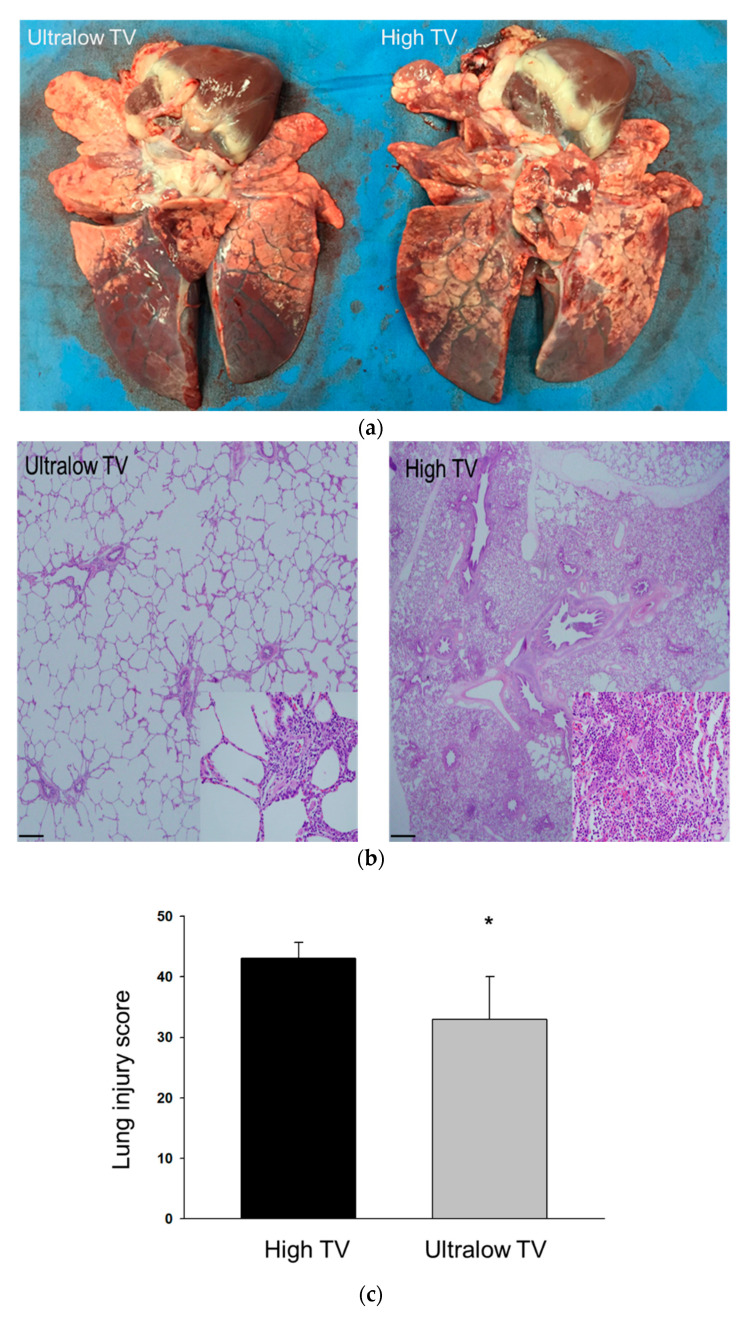
Representative (**a**) gross and (**b**) microscopic images of lung after 12 h of ventilator use in ultralow tidal ventilation group (left panel) and high tidal ventilation group (right panel). (**b**, left panel) Mild perivascular inflammatory cell infiltration and interstitial fibrosis (H&E stain × 10; inset × 200), and (**b**, right panel) more extensive infiltration of inflammatory cells (H&E stain × 40; inset × 200). Scale bars indicate 400 μm (left panel), and 100 μm (right panel). Comparison of (**c**) severity of lung injury assessed by the lung injury score and (**d**) lung wet-to-dry (W/D) weight ratio; * high-tidal-volume ventilation versus ultralow tidal ventilation under ECMO support, *p* < 0.05.

**Table 1 membranes-10-00379-t001:** Vital sign and laboratory variables.

		High Tidal Volume (*n* = 4)	Ultralow Tidal Volume (*n* = 4)	*p*
Baseline	MAP (mmHg)	80.2 ± 13.3	69.2 ± 2.8	0.197
	HR (beats/min)	82.0 ± 7.8	104.8 ± 14.7	**0.034**
	WBC (10^3^/m^3^)	13.8 ± 6.7	15.1 ± 2.7	0.671
	Temperature (°C)	36.3 ± 0.4	36.0 ± 0.3	0.349
1 h	MAP (mmHg)	68.0 ± 7.0	75.5 ± 8.7	0.227
	HR (beats/min)	153.6 ± 121.6	183.1 ± 164.6	0.585
	WBC (10^3^/m^3^)	10.8 ± 5.2	10.2 ± 2.3	0.878
	Temperature (°C)	35.0 ± 0.4	34.0 ± 0.3	0.493
6 h	MAP (mmHg)	47.0 ± 6.2	54.0 ± 6.8	0.177
	HR (beats/min)	76.0 ± 16.3	77.2 ± 7.4	0.893
	WBC (10^3^/m^3^)	4.7 ± 3.2	7.6 ± 2.1	0.189
	Temperature (°C)	33.65 ± 0.1	32.5 ± 0.6	0.118
12 h	MAP (mmHg)	41.0 ± 7.8	36.7 ± 10.7	0.601
	HR (beats/min)	84.8 ± 23.6	65.5 ± 8.6	0.176
	WBC (10^3^/m^3^)	4.82 ± 2.1	5.4 ± 3.2	0.789
	Temperature (°C)	32.7 ± 0.2	30.0 ± 0.9	0.051

Values are expressed as mean ± standard deviation; significant P values are in bold. ECMO: Extracorporeal membrane oxygenation, PaCO_2_: Partial pressure of carbon dioxide, PaO_2_: Partial pressure of oxygen, MV: Minute ventilation, PIP: Peak inspiratory pressure.

**Table 2 membranes-10-00379-t002:** Oxygenation and ventilation profiles.

		High Tidal Volume (*n* = 4)	Ultralow Tidal Volume (*n* = 4)	*p*
Baseline	PaCO_2_ (mmHg)	35.9 ± 9.3	40.8 ± 16.8	0.633
	PaO_2_ (mmHg)	87.1 ± 20.5	86.0 ± 18.3	0.939
	pH	7.49 ± 0.08	7.48 ± 0.10	0.886
	MV (L/min)	6.0 ± 1.5	6.0 ± 1.7	0.948
	PIP (cmH2O)	23.0 ± 6.6	24.0 ± 4.1	0.812
1 h	PaCO_2_ (mmHg)	28.1 ± 7.2	33.4 ± 11.2	0.463
	PaO_2_ (mmHg)	103.3 ± 30.9	87.0 ± 9.4	0.348
	pH	7.53 ± 0.08	7.48 ± 0.09	0.458
	MV (L/min)	8.6 ± 1.1	3.7 ± 2.6	0.014
	PIP (cmH2O)	32.5 ± 5.9	16.5 ± 7.1	0.014
6 h	PaCO_2_ (mmHg)	24.4 ± 3.9	27.4 ± 1.8	0.207
	PaO_2_ (mmHg)	99.2 ± 28.0	78.7 ± 10.1	0.218
	pH	7.55 ± 0.04	7.54 ± 0.02	0.508
	MV (L/min)	7.8 ± 0.7	2.6 ± 0.4	0.001
	PIP (cmH2O)	35.0 ± 4.6	20.0 ± 4.6	0.016
12 h	PaCO_2_ (mmHg)	32.5 ± 8.9	29.9 ± 7.0	0.666
	PaO_2_ (mmHg)	63.9 ± 2.9	94.3 ± 24.6	0.089
	pH	7.48 ± 0.09	7.49 ± 0.08	0.849
	MV (L/min)	6.2 ± 2.0	3.4 ± 1.1	0.046
	PIP (cmH2O)	38.8 ± 3.3	25.2 ± 8.0	0.021

Values are expressed as mean ± standard deviation; significant P values are in bold. ECMO: Extracorporeal membrane oxygenation, PaCO_2_: Partial pressure of carbon dioxide, PaO_2_: Partial pressure of oxygen, MV: Minute ventilation, PIP: Peak inspiratory pressure.

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
