# Peer review of "Effects of Ultralow-Tidal-Volume Ventilation under Veno-Venous Extracorporeal Membrane Oxygenation in a Porcine Model with Ventilator-Induced Lung Injury"

_membranes, 2020, doi:10.3390/membranes10120379_

Round 1

Reviewer 1 Report

The authors performed a well control study in piglet with 4 for each group, conventional ventilation vs ultraslow ventilation 3ml/kg. They challenged the previous study of 4 ml/kg, and they made a conclusion the positive outcome.

They only perfumed in the setting of sepsis-related i jury and it persisted about 12 hours only.

The study is well designed and the data were collected according to the clinical requirement. The discussion also covered all possible limitation. 

Congratulation about the authors' effort and study. 

Author Response

The authors would like to thank the editor and the reviewers for their thoughtful and constructive feedback provided regarding our manuscript. We have carefully addressed all the comments. Our revisions reflect all reviewers’ suggestions and comments. Detailed responses to individual reviewers’ comments are given below.

Reviewer #1

Comment 1: The authors performed a well control study in piglet with 4 for each group, conventional ventilation vs ultraslow ventilation 3ml/kg. They challenged the previous study of 4 ml/kg, and they made a conclusion the positive outcome. They only perfumed in the setting of sepsis-related i jury and it persisted about 12 hours only. The study is well designed, and the data were collected according to the clinical requirement. The discussion also covered all possible limitation. Congratulation about the authors' effort and study.

Response: We greatly appreciate the reviewer’s efforts in carefully reviewing our manuscript.

Reviewer 2 Report

Thank you to the authors of the manuscript entitled "Effects of Ultralow Tidal Volume Ventilation Under Veno-Venous Extracorporeal Membrane Oxygenation in a Porcine Model with Acute Respiratory Distress Syndrome" that have put together an interesting experimental study showed that the use of ECMO with ultralow tidal volume ventilation in pigs with ARDS attenuated lung injury from mechanical ventilation and infection. However, there are a few challenges that need to be addressed.

It is appreciable that the manuscript provides:

  • Highlight the novelty of the study findings
  • The body temperature and pH changes.
  • The sedation using Isoflurane has no affect on the study outcome
  • What was the position of the animal and explain that it did not contribute to the severity of the upper lobe?
  • Comment on the considerable high level of IL-6 cytokine at the baseline.
  • Cite the significance of B-line as a measurement in the U/S lung and provide images.
  • Identify the images in Fig 5 (b) and the Y axis in Fig 5 (c) and (d).
  • Several sentences have been structured incorrectly

Author Response

The authors would like to thank the editor and the reviewers for their thoughtful and constructive feedback provided regarding our manuscript. We have carefully addressed all the comments. Our revisions reflect all reviewers’ suggestions and comments. Detailed responses to individual reviewers’ comments are given below.

Reviewer #2

Comment 1: Highlight the novelty of the study findings

Response: We deeply thank the reviewer for their valuable comments. To the best of our knowledge, this is the first experimental study to provide the firm evidence that ultralow tidal volume ventilation with simultaneous ECMO could alleviate microscopic lung injury induced by a mechanical ventilator. The severity of lung injury was also assessed with the non-invasive lung ultrasound in our study. As reviewer suggested, we highlighted the novel points of our study in the discussion section, as follows.

To the best of our knowledge, this is the first experimental study to provide evidence that ultralow tidal volume ventilation with simultaneous ECMO could alleviate microscopic lung injury induced by a mechanical ventilator. Lung ultrasound was also performed to assess the severity of lung injury in this study. The total lung ultrasound score was increased in both groups, showing a trend for more severe injury in high tidal ventilation group; however, statistically significant differences were not observed in the lung injury score according to the regions and total ultrasound score between the two groups. These findings might be attributed to difficulty accessing the ventral region of animals with an ultrasound probe.

Comment 2: The body temperature and pH changes.

Response: As reviewer suggested, we have added the temperature and pH in the table, as follows.

Table 1. Vital sign and laboratory variables.

High Tidal Volume (n = 4)

Ultralow Tidal Volume

(n = 4)

P

Baseline

MAP (mmHg)

80.2 ± 13.3

69.2 ± 2.8

0.197

HR (beats/min) 

82.0 ± 7.8

104.8 ± 14.7

0.034

WBC (103/m3)

13.8 ± 6.7

15.1 ± 2.7

0.671

Temperature (℃)

36.3 ± 0.4

36.0 ± 0.3

0.349

1 h

MAP (mmHg)

68.0 ± 7.0

75.5 ± 8.7

0.227

HR (beats/min) 

153.6 ± 121.6

183.1 ± 164.6

0.585

WBC (103/m3)

10.8 ± 5.2

10.2 ± 2.3

0.878

Temperature (℃)

35.0 ± 0.4

34.0 ± 0.3

0.493

6 h

MAP (mmHg)

47.0 ± 6.2

54.0 ± 6.8

0.177

HR (beats/min) 

76.0 ± 16.3

77.2 ± 7.4

0.893

WBC (103/m3)

4.7 ± 3.2

7.6 ± 2.1

0.189

Temperature (℃)

33.65 ± 0.1

32.5 ± 0.6

0.118

12 h

MAP (mmHg)

41.0 ± 7.8

36.7 ± 10.7

0.601

HR (beats/min) 

84.8 ± 23.6

65.5 ± 8.6

0.176

WBC (103/m3)

4.82 ± 2.1

5.4 ± 3.2

0.789

Temperature (℃)

32.7 ± 0.2

30.0 ± 0.9

0.051

Table 2. Oxygenation and Ventilation profiles

High Tidal Volume

(n = 4)

Ultralow Tidal Volume

(n = 4)

P

Baseline

PaCO2 (mmHg)

35.9 ± 9.3

40.8 ± 16.8

0.633

PaO2 (mmHg) 

87.1 ± 20.5

86.0 ± 18.3

0.939

pH

7.49 ± 0.08

7.48 ± 0.10

0.886

MV (L/min)

 6.0 ± 1.5

 6.0 ± 1.7

0.948

PIP (cmH2O)

23.0 ± 6.6

24.0 ± 4.1

0.812

1 h

PaCO2 (mmHg)

28.1 ± 7.2

33.4 ± 11.2

0.463

PaO2 (mmHg) 

103.3 ± 30.9

87.0 ± 9.4

0.348

pH

7.53 ± 0.08

7.48 ± 0.09

0.458

MV (L/min)

 8.6 ± 1.1

 3.7 ± 2.6

0.014

PIP (cmH2O)

32.5 ± 5.9

16.5 ± 7.1

0.014

6 h

PaCO2 (mmHg)

24.4 ± 3.9

27.4 ± 1.8

0.207

PaO2 (mmHg) 

99.2 ± 28.0

78.7 ± 10.1

0.218

pH

7.55 ± 0.04

7.54 ± 0.02

0.508

MV (L/min)

 7.8 ± 0.7

 2.6 ± 0.4

0.001

PIP (cmH2O)

35.0 ± 4.6

20.0 ± 4.6

0.016

12 h

PaCO2 (mmHg)

32.5 ± 8.9

29.9 ± 7.0

0.666

PaO2 (mmHg) 

63.9 ± 2.9

94.3 ± 24.6

0.089

pH

7.48 ± 0.09

7.49 ± 0.08

0.849

MV (L/min)

 6.2 ± 2.0

 3.4 ± 1.1

0.046

PIP (cmH2O)

38.8 ± 3.3

25.2 ± 8.0

0.021

Comment 3: The sedation using Isoflurane has no affect on the study outcome

Response: Recent studies reported that the volatile anesthetics such as isoflurane protect the central nervous, renal, and hepatic system from inflammatory injury. However, the use of isoflurane in patients with lung injury remains controversial and the mechanism of its protective effects remains unclear1,2. Moreover, isoflurane was used in both high tidal ventilation and ultralow tidal ventilation group, which had little effect on our results.  

Reference

#1. Hung, C. J. et al. Assessing transient pulmonary injury induced by volatile anesthetics by increased lung uptake of technetium-99m hexamethylpropylene amine oxime. LUNG 181, 1–7 (2003).

#2. Englert, J. A. et al. Isoflurane ameliorates acute lung injury by preserving epithelial tight junction integrity. Anesthesiology 123, 377–388 (2015).

Comment 4: What was the position of the animal and explain that it did not contribute to the severity of the upper lobe?

Response: Animals were maintained in the supine position in our study. As reviewer pointed out, lower and middle lobe, dependent area during supine position, had more severe injury compared to the upper lobe, and severity of injury of these areas was comparable between high tidal ventilation and ultralow tidal ventilation group. However, more severe lung injury was observed in the upper lobe of pigs with high tidal ventilation, compared to those with ultralow tidal ventilation. Consistent with our results, earliest abnormalities appear first in dependent areas in ARDS experimental models, thus non-dependent areas may be more useful for tracking progression and grading ARDS severity while in the supine position1.

Reference

#1. See, K. C., Ong, V., Tan, Y. L., Sahagun, J. & Taculod, J. Chest radiography versus lung ultrasound for identification of acute respiratory distress syndrome: a retrospective observational study. Critical Care 22, 203 (2018).

Comment 5: Comment on the considerable high level of IL-6 cytokine at the baseline.

Response: IL-6 levels of both groups at the baseline were 32.3 and 36.0 pg/ml in our study. Based on the ELISA kits (porcine-specific ELISA kits; R&D Systems Inc., Minneapolis, MN, USA) we used for the IL-6 measurement, baseline values are within normal range of 0-200 pg/ml1,2. However, IL-6 level was not elevated after lung injury in both groups. Because IL-6 response occurs relatively later than the other cytokines3, longer time than 12 hours after injury may be needed for the significant elevation of IL-6.

Reference

#1. Pischke, S. E. et al. Sepsis causes right ventricular myocardial inflammation independent of pulmonary hypertension in a porcine sepsis model. PLoS One, e0218624 (2019).

#2. Gozdzik, W. et al. Beneficial effects of inhaled nitric oxide with intravenous steroid in an ischemia–reperfusion model involving aortic clamping. Int J Immunopathol Pharmacol 31, (2018).

#3. Blackwell, T. S. et al. Multiorgan nuclear factor kappa B activation in a transgenic mouse model of systemic inflammation. Am J Respir Crit Care Med 162, 1095–1101 (2000).

Comment 6: Cite the significance of B-line as a measurement in the U/S lung and provide images.

Response: As reviewer suggested, we have stated the significance of B-lines in lung ultrasound in the discussion section with reference as follows. We have also provided the representative ultrasound images in the figure 4(b).

Lung ultrasound was also performed to assess the severity of lung injury in this study. Its use for evaluating the severity of lung injury has been validated in not only in critically ill patients with ARDS, but in experimental piglet model[20-23]. In this model, B‐lines were correlated with increasing extravascular lung water, and it was detectable before the onset of functional impairments such as hypoxemia[23].

  1. Hew, M.; Tay, T.R. The efficacy of bedside chest ultrasound: from accuracy to outcomes. European Respiratory Review 2016, 25, 230-246, doi:10.1183/16000617.0047-2016.
  2. Koenig, S.; Mayo, P.; Volpicelli, G.; Millington, S.J. Lung Ultrasound Scanning for Respiratory Failure in Acutely Ill Patients: A Review. Chest 2020, 10.1016/j.chest.2020.08.2052, doi:10.1016/j.chest.2020.08.2052.
  3. Elsayed, Y.N.; Hinton, M.; Graham, R.; Dakshinamurti, S. Lung ultrasound predicts histological lung injury in a neonatal model of acute respiratory distress syndrome. Pediatric Pulmonology 2020, 55, 2913-2923, doi:https://doi.org/10.1002/ppul.24993.
  4. Gargani, L.; Lionetti, V.; Di Cristofano, C.; Bevilacqua, G.; Recchia, F.A.; Picano, E. Early detection of acute lung injury uncoupled to hypoxemia in pigs using ultrasound lung comets. Crit Care Med 2007, 35, 2769-2774, doi:10.1097/01.Ccm.0000287525.03140.3f.

Comment 7: Identify the images in Fig 5 (b) and the Y axis in Fig 5 (c) and (d).

Response: As reviewer suggested, we have identified figure 5 (b) and the Y axis in figure 5(c) and (d) as follows.

Comment 8: Several sentences have been structured incorrectly

Response: We have proofread the manuscript again to correct the English grammar and it has also been thoroughly reviewed by a specialized English language editing service. We have attached the certification from the English language editing service. We hope that the current version of the manuscript has been improved in terms of grammar, language, and overall readability.